# Prompting Multilingual Large Language Models to Generate Code-Mixed Texts: The Case of South East Asian Languages

**Zheng-Xin Yong**[1]  **Ruochen Zhang**[1]  **Jessica Zosa Forde**[1]  **Skyler Wang**[2]
**Arjun Subramonian**[3]  **Holy Lovenia**[4]  **Samuel Cahyawijaya**[4]  **Genta Indra Winata**[5]
**Lintang Sutawika**[6]  **Jan Christian Blaise Cruz**[7]  **Yin Lin Tan**[8,9]  **Long Phan**[10]
**Rowena Garcia**[11]  **Thamar Solorio**[12]  **Alham Fikri Aji**[12]

[1]Brown University  [2]UC Berkeley  [3]University of California, Los Angeles
[4]HKUST  [5]Bloomberg  [6]EleutherAI  [7]Samsung R&D Institute Philippines
[8]Stanford University  [9]National University of Singapore  [10]VietAI Research
[11]University of Potsdam  [12]MBZUAI

## Abstract

While code-mixing is a common linguistic practice in many parts of the world, collecting high-quality and low-cost code-mixed data remains a challenge for natural language processing (NLP) research. The recent proliferation of Large Language Models (LLMs) compels one to ask: how capable are these systems in generating code-mixed data? In this paper, we explore prompting multilingual LLMs in a zero-shot manner to generate code-mixed data for seven languages in South East Asia (SEA), namely Indonesian, Malay, Chinese, Tagalog, Vietnamese, Tamil, and Singlish. We find that publicly available multilingual instruction-tuned models such as BLOOMZ and Flan-T5-XXL are incapable of producing texts with phrases or clauses from different languages. ChatGPT exhibits inconsistent capabilities in generating code-mixed texts, wherein its performance varies depending on the prompt template and language pairing. For instance, ChatGPT generates fluent and natural Singlish texts (an English-based creole spoken in Singapore), but for English-Tamil language pair, the system mostly produces grammatically incorrect or semantically meaningless utterances. Furthermore, it may erroneously introduce languages not specified in the prompt. Based on our investigation, existing multilingual LLMs exhibit a wide range of proficiency in code-mixed data generation for SEA languages. As such, we advise against using LLMs in this context without extensive human checks.

## 1 Introduction

Code-mixing, also known as code-switching, is the linguistic practice of alternating between two or more languages in an utterance or conversation (Poplack, 1978). It allows individuals to express culturally-specific ideas, connect with or differentiate from other interlocutors, and reify their identities (Bhatia and Ritchie, 2004; Grosjean,

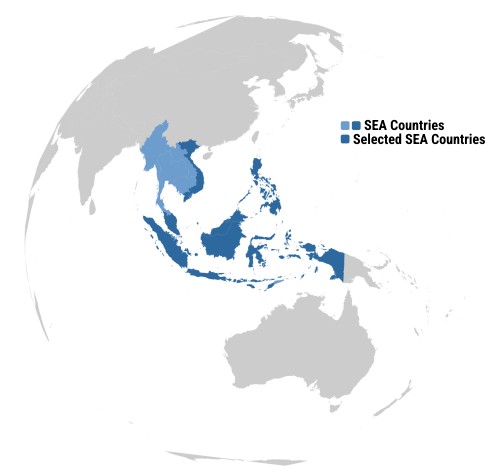

Figure 1: Depiction of SEA regions, which consist of a total of 11 countries. We prompt LLMs to generate code-mixed data of languages used in six South East Asian countries (colored in dark blue): Brunei, Indonesia, Malaysia, Philippines, Singapore, and Vietnam.

1982; Toribio, 2002; Chen, 1996; Doğruöz et al., 2021). Despite its prevalence across many parts of the world, computational research into this area remains understudied (Diab et al., 2014; Aguilar et al., 2020; Winata et al., 2021, 2022; Zhang et al., 2023).

One longstanding challenge in this area involves acquiring high-quality and low-cost code-mixed data. For one, code-mixing is observed more frequently in colloquial settings and spoken communication, which makes procuring and curating extensive datasets logistically demanding and costly (Chan et al., 2009; Winata et al., 2021). Moreover, despite code-mixing's prevalence across social media and digital messaging platforms, consolidating such data may be curtailed by legal guardrails and scalability issues. Recognizing these challenges, we explore the feasibility of using generative Large Language Models (LLMs) to ameliorate data scarcity in code-mixing research. As

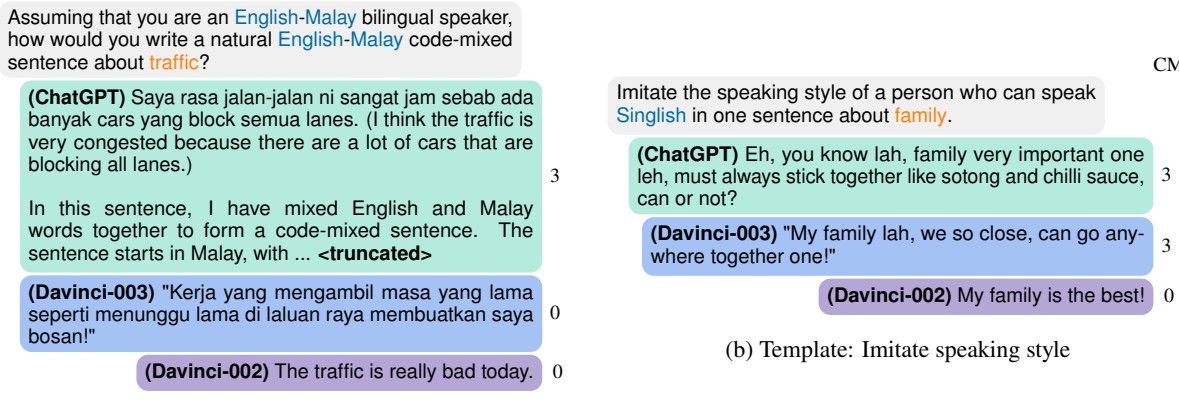

Figure 2: Example prompt templates with different languages and topic fields and responses from LLMs containing code-mixed / non-code-mixed sentences. Note that the explanations are a part of ChatGPT's original generation. "CM" indicates the level of code-mixing (Section 2.2). See Figure 15 in Appendix for all prompt templates and responses from other LLMs such as BLOOMZ and Flan-T5-XXL.

recent work shows that LLMs can successfully generate synthetic data (Taori et al., 2023; He et al., 2023; Tang et al., 2023; Whitehouse et al., 2023), here we evaluate whether multilingual LLMs can be prompted to create code-mixed data that look natural to native speakers (and if so, to what extent).

To this end, we hone in on languages in South East Asia (SEA). Home to more than 680 million people and over 1200 languages, code-mixing is particularly prevalent in this region due to its countries' extended histories of language and cultural cross-fertilization and colonialism (Figure 1) (Goddard, 2005; Bautista and Gonzalez, 2006; Reid et al., 2022). Marked by its distinctive multilingual and multiracial composition today, SEA presents an opportunity to further research numerous marginalized languages and linguistic practices in NLP research[1] (Migliazza, 1996; Goddard, 2005; Joshi et al., 2020; Aji et al., 2022; Winata et al., 2023; Cahyawijaya et al., 2022). Nonetheless, publicly available code-mixed datasets relevant to SEA communities remain limited (Lyu et al., 2010; Winata et al., 2022).

We prompt five multilingual LLMs, i.e., ChatGPT, InstructGPT (davinci-002 and davinci-003) (Ouyang et al., 2022), BLOOMZ (Muennighoff et al., 2022), and Flan-T5-XXL (Chung et al., 2022) to generate code-mixed text that bilingually

mixes English with either **Malay, Indonesian, Chinese, Tagalog, Vietnamese, or Tamil**. All of these six SEA languages (alongside English) are used across six SEA countries, namely Singapore, Malaysia, Brunei, Philippines, Indonesia, and Vietnam. Furthermore, they belong to different language families—Indo-European, Austronesian, Sino-Tibetan, Austro-Asiatic, and Dravidian. An example of a prompt we used is: "Write an English and Tamil code-mixed sentence about Artificial Intelligence." In addition, we prompt these LLMs to generate texts in **Singlish**, an English-based creole widely spoken in Singapore that combines multiple SEA languages such as Malay, Chinese and Tamil. We ask native speakers to annotate the *naturalness* (i.e., whether a native speaker would speak as such) and the *level of code-mixing* in the outputs.

To the best of our knowledge, our work marks the first attempt at studying the generation of synthetic code-mixed data through prompting LLMs in a zero-shot fashion without any monolingual reference texts or explicit linguistic constraints (Solorio and Liu, 2008; Tarunesh et al., 2021; Rizvi et al., 2021; Mondal et al., 2022). We find that publicly available multilingual language models such as BLOOMZ and Flan-T5-XXL are only capable of code-mixing with loanwords or topic-related nouns. Most of the time, they fail to code-mix (despite being advertised as multilingual). While ChatGPT stands out in its ability to generate code-mixed texts, it is extremely sensitive to the prompt template and exhibits a considerable variance of success in generating natural-sounding code-mixed

---

[1]Major languages in SEA countries belong to different language families such as Indo-European, Thai, Austronesian, Sino-Tibetan, Dravidian, and Austro-Asiatic. Furthermore, there are at least thousands of major and minor SEA languages.

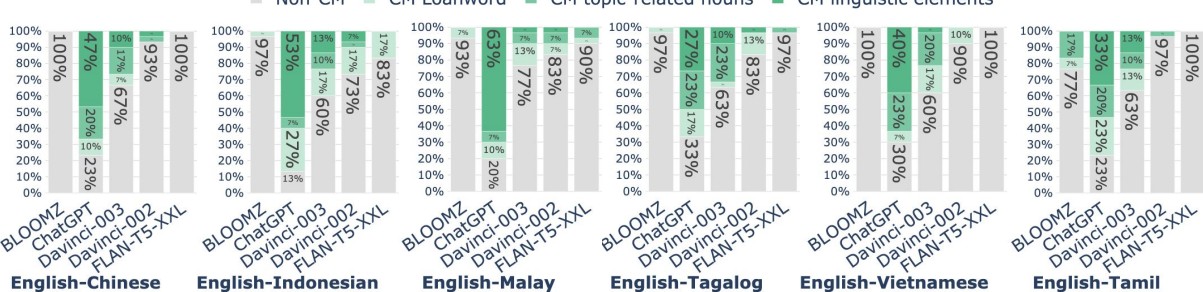

Figure 3: Comparison of performance of different LLMs in generating code-mixed data through zero-shot prompting. We distribute the result across different code-mixing levels: (0) No code-mixing (Non-CM), (1) Loanword, (2) Topic-related nouns, and (3) Linguistic Elements.

texts across different language pairs. Additionally, it may erroneously introduce additional languages not specified in the prompt and wrongly explain the code-mixing of the text.

Our results lead us to conclude that code-mixing, at least as of today, is not considered an essential component of many multilingual LLMs. Moreover, the opaque creation of models like ChatGPT makes it difficult to ascertain the mechanisms that enable code-mixing. By highlighting the limited promises of LLMs in a specific form of low-resource data generation, we advise NLP researchers against using existing systems to produce synthetic code-mixed data without extensive human evaluation.

## 2 Methodology

### 2.1 Prompting Language Models

We collect synthetic code-mixed data by prompting LLMs with requests along two axes: languages and topics (food, family, traffic, Artificial Intelligence, and weather). See Figure 2 for examples of different prompt templates. Specifically, we explore ChatGPT, InstructGPT (davinci-002 and davinci-003) (Ouyang et al., 2022), 176B-parameter BLOOMZ (Muennighoff et al., 2022), and Flan-T5-XXL (Chung et al., 2022). We use OpenAI and HuggingFace's API for prompting (see Appendix B), except in the case of ChatGPT, which we manually queried through its web interface[2].

In our prompts, we specify code-mixing between English and either Indonesian, Malay, Mandarin, Tagalog, Vietnamese, or Tamil. We focused on code-mixing English with SEA languages for two reasons: (1) extensive literature on code-mixed English provides a relevant point of comparison,

and (2) English is one of the most widely used languages in code-mixing across SEA countries (Kirkpatrick, 2014). We additionally prompt with sentences in Singlish, a creole language, to evaluate how sensitive LLMs are to the diversity of language practices in the SEA region. In total, we submitted 210 unique prompts per language model.

### 2.2 Evaluation

**Level of Code-Mixing**

To evaluate outputs, we ask whether LLMs can produce *intrasentential* code-mixed text. We adopt the definition of intrasentential code-mixing from Berk-Seligson (1986), which covers mixing small constituents—such as noun and verb phrases—and large constituents—such as coordinate clauses and prepositional phrases. Native speakers are then tasked to manually annotate the collected responses on a scale from 0 to 3 using the following coding guidelines to denote the degree of code-mixedness:

- **0 - No code-mixing:** The generated text is written purely in one language or only exhibits *intersentential* code-mixing (i.e., switching at sentence boundaries including interjection, idiom, and tags). We adopt the definition from Berk-Seligson (1986).
- **1 - Loanwords:** The generated text uses loanwords for common terminologies. We consider a word as a loanword if it is listed in Wiktionary[3]. For example: In the sentence, "I like eating *pho*," "pho" is a loanword.
- **2 - Topic-related nouns:** The generated text uses nouns related to the topic specified in the prompt in another language. For instance, for the topic of traffic, an example would be "今天的 *traffic* 真的很糟糕，我开了一个小时

---

[2]ChatGPT's API was not publicly released when we conducted this study.

[3]https://en.wiktionary.org

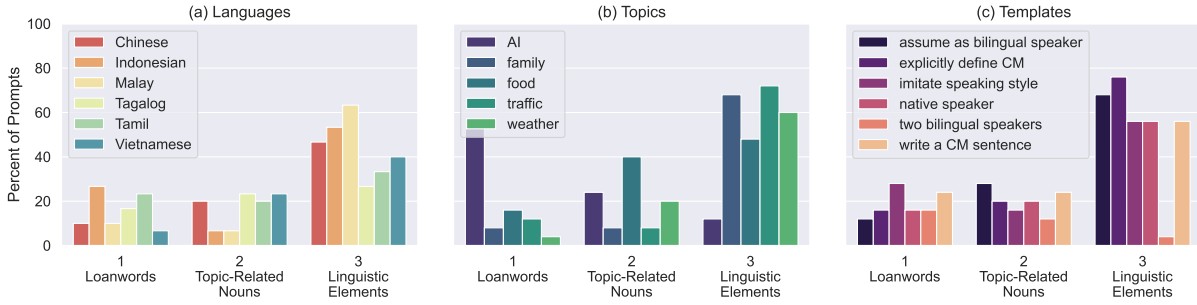

Figure 4: Analysis of code-mixed data generated by ChatGPT.

才到了办公室。" (Chinese: "The traffic today is really terrible. I spent an hour driving to get to the office.")

- **3 - Linguistic Elements:** The generated text mixes linguistic elements beyond loanwords and topic-related nouns at the phrasal or clausal level. One example is verb phrases: "My family *ay nagplano ng isang malaking* family reunion *sa park* this coming weekend." (Tagalog: "My family has planned a big family reunion at the park this coming weekend.") This category also includes intraword code-mixing, e.g., "Kapag busy ang trapiko, mag-ingat ka sa *pagda-drive*[4] para maiwasan mo ang mga masamang pangyayari." (Tagalog: "When traffic is busy, be careful while driving to avoid accidents.")

We use this scale instead of popular word-level metrics such as CMI (Gambäck and Das, 2014) because our scale more holistically evaluates the ability of LLMs to code-mix. The lower end of this scale reflects a lower complexity of code-mixing. Code-mixing with loanwords is arguably less challenging, as they are often used in a monolingual context to begin with. Likewise, code-mixing topic-related nouns is not as complex as there is presumably a correspondence between the nouns in the two languages and is primed by the prompts.

On the other hand, code-mixing prefixes/suffixes, phrases and clauses requires a good grasp of the intricate morphosyntactic structures of both languages and can produce syntactically diverse code-mixed data. Therefore, we consider the LLM to have *successfully* generated code-mixed text only if the text belongs to this category.

---

[4]The prefix "pag-" in Tagalog is affixed to the English word "drive", resulting in the word "pagda-drive" (the act of driving). This example demonstrates the application of Tagalog inflection rules to English words.

**Naturalness**

We asked native speakers to annotate the naturalness of the generated text on a rating scale of 1—not natural at all, 2—an annotator may see someone else from other regions/cultures speaking it, and 3—an annotator may see themselves speaking it. If the text is not considered natural (rating of 1), we requested the annotators to document their reasons, including fluency issues, syntactical errors, and semantic incorrectness.

Code-mixing practices vary across race, gender, class, and geography, among other axes. Therefore, in line with sociolinguistic research, we emphasize that the "naturalness" of code-mixed text (as perceived by an annotator) is subjective, and depends on the background and social location of the annotator. All annotators are AI researchers and native speakers of their respective SEA languages. We refer to Appendix G for more information about our annotators and their inter-annotator agreement scores.

**Accurateness**

We additionally annotate the *accurateness* of the generations to account for task failure or incorrect explanations. Inaccurate outputs were defined by:

- **Failure to follow instructions:** Some LLMs fail to accurately carry out the prompt instruction (for example, generating monolingual sentences). We also observe some code-mixed outputs that include additional languages unspecified in the prompts.
- **Inaccurate explanations:** ChatGPT, among other LLMs, tends to explain the code-mixed phrases, but occasionally provides incorrect explanations (see Figure 10).

> Imagine there are two English-Vietnamese bilingual speakers talking to each other, can you imagine a conversation between them talking about family?

> **(ChatGPT)** Person A: Hi, how's your family doing?
> Person B: Chào bạn, gia đình tôi đang khỏe. Còn gia đình bạn thì sao?
> Person A: They're good, thanks for asking. How about yours?
> Person B: Gia đình tôi cũng khỏe, cảm ơn bạn. Mới đây, tôi đã đi chơi với ông bà và các em họ của mình.

> Imagine there are two English-Chinese bilingual speakers talking to each other, can you imagine a conversation between them talking about weather?

> **(ChatGPT)** English Speaker 1: Hey, have you been outside today? It's really hot!
> Chinese Speaker 1: 是啊，今天天气真的很热。
> English Speaker 2: Yeah, it's been pretty sunny all day.
> Chinese Speaker 2: 是啊，今天一整天都很晴朗。

Figure 5: Failure cases when we prompt ChatGPT to imagine two bilingual speakers conversing about a specific topic.

## 3 Results

### 3.1 English-SEA Languages

Figure 3 demonstrates that ChatGPT outperforms other language models in generating code-mixed data across six different language pairs, and is particularly proficient at code-mixing linguistic expressions. For InstructGPT, the davinci-003 model demonstrates some ability to generate syntactically varied code-mixed texts, whereas davinci-002 mostly code-mixes with loanwords. Conversely, the other two publicly available multilingual LLMs have extremely limited code-mixing capabilities. More specifically, Flan-T5-XXL can only code-mix with loanwords for the Indonesian-English language pair, and most of its non-English monolingual outputs suffer from severe fluency issues (see Appendix D). BLOOMZ is only capable of code-mixing topic-related nouns for Tamil-English even though its multilingual pretraining data ROOTS (Laurençon et al., 2022) and instruction-tuning data xP3 (Muennighoff et al., 2022) cover Indonesian, Chinese, Tamil, and Vietnamese. We observe no direct effects of the proportions of these languages in the training sets on BLOOMZ's ability to code-mix (Appendix E).

We further break down the performance of Chat-GPT in Figure 4[5]. In Figure 4(a), we see that

---

[5] Detailed analysis for davinci-002, davinci-003, Flan-T5-XXL and BLOOMZ can be found in the Appendix (Figure 11, Figure 12, Figure 13 and Figure 14).

ChatGPT is least proficient at mixing linguistic elements for English-Tagalog. This may be due to syntactic differences between the two languages; for example, English exhibits Subject-Verb-Object (SVO) word order, whereas Tagalog exhibits a verb-initial structure. Moreover, English demonstrates nominative-accusative alignment, whereas Tagalog, being a symmetrical-voice language, utilizes a case system with a typological classification that "remains controversial among Austronesian linguists" (Aldridge, 2012, 192). In contrast, ChatGPT performs the best for English-Indonesian code-mixing, which may be due to training data distribution and similarities between the two languages regarding word order and morphosyntactic alignment. We also find that ChatGPT is capable of using either English or a SEA language as the matrix language, i.e., as the main language of a sentence as per the Matrix Language Frame model (Myers-Scotton, 1997).

Figure 4(b) shows ChatGPT's code-mixing proficiency based on topics. ChatGPT tends to code-mix with loanwords when the topic is about "AI" by mixing the English loanwords "Artificial Intelligence," or its short form "AI." For food, it tends to code-mix with food-related terms—which are topic-related nouns—in SEA languages such as "bánh mì" (Vietnamese sandwich). We also observe some representation biases in specific language-topic pairs. For instance, when it comes to food, ChatGPT uses the word "nasi goreng" (fried rice) for all English-Indonesian responses. For other topics, such as traffic and weather, it tends to code-mix phrases related to traffic congestion and hot weather.

In Figure 4(c), we find the prompt template with the highest quality results is the one where the term code-mixing is explicitly defined. In contrast, the worst-performing template consists of asking the model to generate conversations between two bilingual speakers, where the term code-mixing is unmentioned. In Figure 5, we see that ChatGPT generates an uncommon pattern of conversations where one interlocutor speaks in English and the other speaks in another language entirely (top example). Furthermore, ChatGPT may assume there are four speakers though the prompt asks for a conversation between two speakers (bottom example).

In terms of naturalness, we observe a considerable variance in ChatGPT's ouputs, with English-Tamil being the least natural (Figure 6). Further

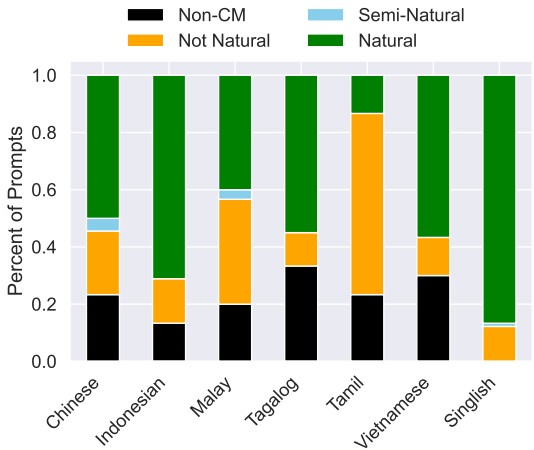

Figure 6: Naturalness of code-mixed (CM) text generated from ChatGPT. "Semi-Natural" indicates that the annotators see someone else from other region/culture speaking it, whereas "Natural" indicates that the annotators see themselves speaking it.

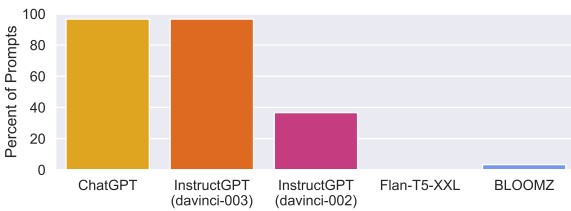

Figure 7: Models' performance in generating Singlish expressions through zero-shot prompting.

analysis shows that ChatGPT either commits grammatical mistakes (such as comma splice and redundancy) or generates semantically confusing sentences. We also observe unnatural text patterns that mix two different script systems of the same language in the single sentence (for example, the Tamil script and its transliterated Latin script). We document these naturalness and fluency issues in Table 4 (Appendix F) and report the interannotator agreement scores in Appendix G.

### 3.2 Singlish

> Imagine that you are a Singaporean speaker. How would you talk about food using Singlish in a single sentence?

> **(ChatGPT)** "Wah, the hawker center got so many sedap food, like chicken rice, laksa, and rojak, can die for sia!" (Translation: "Wow, the hawker center has so many delicious foods, like chicken rice, laksa, and rojak, they are to die for!")

Figure 8: Singlish sentence and its translation to Standard English, both generated by ChatGPT.

Given that Singlish is an English-based creole language that has lexicalized a lot of words from many languages such as Malay, Chinese, Tamil, Cantonese, and Hokkien (e.g., "lah," "leh," and "loh," etc.), it is difficult to disambiguate loanwords and linguistic elements. Therefore, we classify the model success based on a binary scale of whether the generated sentences contain Singlish expressions (e.g., interjections, lexicalized phrases of languages without a standardized writing system)[6]. In Figure 7, we see that ChatGPT and InstructGPT (davinci-003) have up to a 96% success rate in generating Singlish sentences, whereas Flan-T5-XXL and BLOOMZ have a near-zero success rate. Furthermore, we find that ChatGPT is also capable of translating Singlish expressions into Standard American English (Figure 8).

While ChatGPT is capable of generating fluent and natural Singlish (Figure 6), we observe semantic inaccuracies in their word choices that are discernible by native speakers. In Figure 2(b), ChatGPT generates an English-Malay code-mixed phrase "sotong and chilli sauce" where "sotong" is a Malay word for "squid." Upon first inspection, the sentence appears syntactically sound but native speakers found the analogy between familial connections and the dish (squid with chili sauce) to be semantically confusing.

OpenAI's documentation of model differences[7]—more specifically, that davinci-003 can produce higher-quality writing, process more complex instructions, and generate longer content than davinci-002—does not adequately explain why ChatGPT and davinci-003 significantly outperform davinci-002 and other multilingual LLMs in generating Singlish text. We hypothesize that the performance gap may be due to the larger presence of Singlish in OpenAI's training data. Singlish is not only one of the most well-researched dialects of English (Sin, 2017), but it also uses the English alphabet and Latin script systems (unlike many other SEA languages), which potentially makes it easier for LLMs to scrape and parse data in this language.

### 3.3 Failures of ChatGPT

We notice that ChatGPT may fail to produce correctly code-mixed text when the nationality of the speaker is mentioned. Invoking nationality may

---

[6]A reference list can be found at https://en.wikipedia.org/wiki/Singlish_vocabulary

[7]https://help.openai.com/en/articles/6779149-how-do-text-davinci-002-and-text-davinci-003-differ

> Imagine that you are a Malaysian speaker. How would you talk about traffic using both English and Chinese in a single sentence?

> **(ChatGPT)** Assuming that you are asking for a sentence that combines both English and Chinese in a natural and commonly used way by Malaysian speakers, here is an example:
> "今天的 traffic 很 teruk，我要赶紧走去 KLCC for meeting。"
> This sentence incorporates the English word "traffic" and the Chinese word "teruk" (which means "bad" or "terrible" in English) to describe the situation of the traffic being very bad. The speaker then switches back to English to express their intention to hurry to KLCC for a meeting. This kind of code-switching is common in Malaysia, where speakers often mix languages in their daily conversations.

Figure 9: ChatGPT's failure to follow the prompt. Underlined texts show its language misclassification.

induce ChatGPT to generate code-mixed texts with an *additional* national language of the country. In the Figure 9 example, because the national language of Malaysia is Malay[8], ChatGPT not only returns a code-mixed response that includes the Malay word "teruk" when prompted for English-Chinese utterance by a Malaysian, but also misclassifies the word "teruk" as Chinese.

While ChatGPT can explain how its generated text is code-mixed (as illustrated in Figure 2(a)), we find its justifications unreliable for two reasons. First, it may hallucinate words that do not exist in the code-mixed text. For instance, in the top example in Figure 10, the English word "cold" does not exist in the code-mixed text. Second, it may assign words or phrases to the wrong language. In the bottom example in Figure 10, the phrase "nak minum air sejuk je" is in Malay, not English. This observation corroborates Bang et al.'s (2023) finding suggesting ChatGPT performs poorly on language identification tasks for low-resource languages.

## 4 Implications for LLMs and Ways Forward

In this section, we walk through various implications of our work for understanding the deployment of LLMs for synthetic data generation in low-resource settings, multilinguality in LLMs, importance of code-mixing ability in generative LLMs, and research transparency.

**Deploying LLMs for Low-Resourced Data Generation** By putting LLMs' generative capabilities to the test, we ask in this work if they can generate high-quality and low-cost code-mixed texts for

---

[8]Mandarin Chinese is widely spoken in the country.

---

> Imagine that you are an Indonesian speaker. How would you talk about weather using both English and Indonesian in a single sentence?

> **(ChatGPT)** If I were to talk about weather using both English and Indonesian in a single sentence, I might say something like: "*Hari ini sangat hot dan panas sekali, saya rasa aku butuh minum air dingin.*" This sentence includes the English words "hot" and "cold" to describe the weather, and the Indonesian words "panas" and "dingin" which have similar meanings. Additionally, the sentence also includes a personal statement in Indonesian, "saya rasa aku butuh minum air dingin," which means "I feel like I need to drink cold water."

> Code-mixing refers to a phenomenon of combining two or more languages in a single sentence. Can you generate a code-mixed English-Malay sentence about weather?

> **(ChatGPT)** "*The weather hari ni sangat panas, I rasa macam nak minum air sejuk je.*" (Translation: "The weather today is very hot, I feel like drinking cold water.")
> In this sentence, "hari ni" and "sangat panas" are in Malay, while "I" and "nak minum air sejuk je" are in English.

Figure 10: Underlined texts show wrong explanations about the code-mixed text. We italicize the *code-mixed sentences* to make it explicit to the reader.

researchers working on a topic plagued by limited data availability. While we conclude that ChatGPT has shown relative success in generating code-mixed texts for some SEA languages, we advise researchers to exercise heavy caution when using this data generation technique. Even for Singlish, which outperforms the other languages examined, we find that syntactically-sound responses may contain semantic inaccuracies that are difficult for non-native speakers to detect. Furthermore, its explanations may be misleading. Due to the lack of reliability, we strongly suggest researchers to implement extensive human checks with native speakers if they wish to pursue this method of data generation.

**Multilingual $\neq$ Code-Mix Compatible** Our results with BLOOMZ and Flan-T5-XXL show that the ability to code-mix is not acquired by LLMs after pretraining and/or finetuning with multilingual data (Laurençon et al., 2022; Muennighoff et al., 2022; Chung et al., 2022). In other words, for most NLP models, multilinguality simply means that the same system can process tasks and generate outputs in multiple languages, but not necessarily in the same sentence. By highlighting this limitation, we echo previous research motivating the inclusion of code-mixing abilities in NLP models. Doing so requires NLP models to capture the dynamics of combining languages that have different degrees

of typological affinities, as well as pragmatic and contextual features such as tone, formality, and other cultural nuances (Winata et al., 2020; Lai and Nissim, 2022; Kabra et al., 2023).

**Towards More Inclusive Language Technology**
Recognizing that generative LLMs are the primary driving force behind the advancement of AI conversational agents and speech technology (Thoppilan et al., 2022; SambaNova Systems, 2023; Pratap et al., 2023), we emphasize the significance of incorporating code-mixed output recognition and generation capabilities in LLMs in order to enhance the inclusivity and humaneness of language technology. By enabling conversational agents to reflect the language-mixing patterns of the users, people can communicate in ways that are more comfortable and authentic to their linguistic identities. In fact, a recent study by Bawa et al. (2020) has shown that multilingual users strongly prefer chatbots that can code-mix. Removing the need for people to adjust their speech patterns to become legible to machines would not only mitigate the effects of linguistics profiling (Baugh, 2005; Dingemanse and Liesenfeld, 2022) and hegemonic, Western-centric technological designs, but also enable users to develop more trust with language technology through naturalistic dialogue interactions.

**Research Transparency**   Aside from showing that ChatGPT and InstructGPT *can* code-mix, we cannot confidently identify *how* the models do so due to the lack of transparency in how these systems are developed. Without a window into training data and engineering processes that went into models like ChatGPT, we can only speculate that their training data includes a substantial amount of code-mixed texts. To help facilitate greater levels of transparency and accountability, we urge forthcoming LLMs to be more open about how the models were developed and to document accurately and comprehensively the training data used.

## 5   Related Work

**Code-Mixed Data in SEA**   Unlike monolingual data, there is only a limited number of human-curated code-mixed datasets.   This resource limitation is more severe in SEA due to its marginalization in NLP research (Winata et al., 2022).  Popular current code-mixing evaluation benchmarks (Aguilar et al., 2020; Khanuja et al., 2020) do not include SEA languages, and ex-

isting code-mixing studies in SEA only cover a limited number of language pairs and creoles, e.g., English-Tagalog (Oco and Roxas, 2012), English-Indonesian (Barik et al., 2019; Yulianti et al., 2021), Javanese-Indonesian (Tho et al., 2021), Chinese-English (Lyu et al., 2010; Lovenia et al., 2022; Zhang and Eickhoff, 2023) and Singlish (Chen and Min-Yen, 2015; Lent et al., 2021)[9]. The current corpus does not even scratch the surface of the sheer amount of code-mixedness in SEA (Redmond et al., 2009), where deployable data is practically non-existent. In this work, we try to close this gap by exploring the potential of generating synthetic code-mixed data for the SEA region by prompting LLMs.

**Synthetic Code-Mixing**   Generation of synthetic code-mixed data to address data scarcity problem has been previously explored.  Solorio and Liu (2008), Winata et al. (2019), and Tan and Joty (2021) have attempted to generate synthetic code-mixed sentences through word alignment and candidate selection from a parallel corpus. Liu et al. (2020) and Adilazuarda et al. (2022) have similarly generated synthetic code-mixed sentences by replacing words in monolingual sentences with their machine-translated counterparts, whereas Pratapa et al. (2018), Rizvi et al. (2021) and Santy et al. (2021) leveraged parse tree structure for such replacements. Another approach is to perform neural machine translation to translate monolingual sentences to code-mixed ones (Appicharla et al., 2021; Gautam et al., 2021; Jawahar et al., 2021; Dowlagar and Mamidi, 2021). In this work, we assess a novel way of generating synthetic code-mixed sentences through prompting multilingual LLMs.

## 6   Conclusion

To ameliorate the scarcity of code-mixed data for South East Asian languages, we explore generating synthetic code-mixed data using state-of-the-art multilingual Large Language Models (LLMs). On one hand, we find that publicly available LLMs such as BLOOMZ and Flan-T5-XXL have limited capability in generating syntactically diverse code-mixed data. On the other hand, closed-source models such as ChatGPT and InstructGPT are better at generating natural code-mixed text, but their performance varies substantially depending on the

---

[9]To exacerbate the situation, some of the SEA code-mixed datasets are no longer publicly available.

prompt template and language pairing. Furthermore, many outputs suffer from syntactic, semantic, and reliability issues. Therefore, we caution against using LLM-generated synthetic code-mixed data without the involvement of native speakers for annotating and editing.

# 7 Limitations

## 7.1 Effectiveness of Synthetic Code-Mixed Data on Downstream Tasks

In our study, we did not evaluate how much our synthetically generated code-mixed data improve the ability of language models to handle code-mixed text in downstream NLP tasks. While previous findings have shown that finetuning models with synthetic code-mixed data yields less performance gains than with naturally occurring code-mixed data (Santy et al., 2021), we believe that this performance gap will diminish as the quality of synthetic data generation gets better with future multilingual LLMs.

## 7.2 Lack of Human-Generated Data

While we annotated the degree of code-mixedness and naturalness, we did not have human-generated, naturally occurring, code-mixed sentences in response to the prompt topics. Therefore, we could not systematically compare the data distribution of our synthetic data against the human-generated data. However, since there are multiple ways in which a sentence can be code-mixed, our focus in this work is on how human-like are the sentences, and this, we believe, was adequately captured by our evaluation.

## 7.3 Monolingual Zero-Shot Prompting

Our study only uses prompt templates written in English to prompt language models in a zero-shot manner. In future follow-ups, we will (1) use code-mixed prompt templates such as "Generate an English-Bahasa sentence" instead of "Generate an English-Malay sentence" and (2) investigate LLMs' capabilities in generating code-mixed data with in-context few-shot examples.

## 7.4 Instruction-Tuned Language Models

Our work only covers instruction-tuned language models. In future work, we will include a comparison between multilingual models that are not finetuned with instructions—for example, GPT3 (davinci) (Brown et al., 2020) and BLOOM (Scao

et al., 2022)—to explore the effects of instruction tuning in generating code-mixed data.

## 7.5 English-Centric Code-Mixing

Our study focuses on generating code-mixed data only for English-SEA language pairs. For future studies, we plan to investigate generating code-mixed data for non-English language pairs commonly spoken in SEA countries (such as Malay-Chinese and Indonesian-Javanese).

## 7.6 Failures of BLOOM and Flan-T5-XXL

Given the lack of research transparency on why ChatGPT performs better at code-mixed text generation, we assume that the publicly available models such as BLOOM and Flan-T5-XXL are unable to code-mix due to the lack of code-mixed texts in the pretraining corpora and code-mixing tasks in the instruction-tuning datasets. Further investigation is warranted to understand the effects of code-mixed text in pretraining and instruction-tuning data on code-mixed text generation.

## 7.7 Presence of Synthetic Code-Mixed Data in Future Pretraining Data

As we advocate for the code-mixing ability in future generations of LLMs, we are aware of the potential risks of *data feedback*, where generative models that recursively train on data generated by previous generations may amplify biases and lose information about the tails of the original distribution (Shumailov et al., 2023; Taori and Hashimoto, 2022). Since these negative effects can be mitigated through human-generated content (Shumailov et al., 2023), it becomes imperative for the NLP community to collect natural code-mixed data for low-resource languages.

# 8 Ethical Considerations

Code-mixing reflects the linguistic, social, and cultural identity of a multilingual community. Researchers and practitioners should approach synthetic code-mixing with sensitivity and respect, and be cognizant of the potential risks of cultural appropriation or misrepresentation when generating code-mixed data using LLMs. Since LLMs are trained on web data, they may encode biases perpetuating stereotypes, discrimination, or marginalization of specific languages or communities. Prior work has also documented how synthetic data may play a role in feedback loops that amplify the

presence of biased language generation (Taori and Hashimoto, 2022). Therefore, collaboration with linguists, language experts, and community representatives is necessary to avoid the unintentional perpetuation of stereotypes and cultural insensitivity.

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

# A  Languages Spoken in SEA

There are more than 1,200 languages spoken in SEA (Redmond et al., 2009; Maliwat, 2021), 700 of which are spoken in Indonesia (Aji et al., 2022; Cahyawijaya et al., 2022). We describe the languages the SEA languages used in the study in the following paragraphs.

**Mandarin Chinese**  Mandarin Chinese (zh-Hans), which belongs to the Sino-Tibetan language family and uses the Hanzi script, is widely spoken in SEA due to the migration of Chinese people from the coastal provinces of southeastern China, such as Fujian, Guangdong, and Hainan. People of Chinese heritage in SEA frequently use the term "华人" (huá rén) to express their cultural identity as an ethnic group, instead of "中国人" (zhōng guó rén) which is primarily associated with nationality, even though both terms can be translated as "Chinese (people)." Singapore has the largest Chinese ethnic group among all SEA countries and Mandarin Chinese is considered one of the official languages in Singapore.

The language is characterized as linguistically "isolating" in that each Chinese character corresponds to one morpheme and that the language uses very little grammatical inflection. It uses a logographic writing system, which uses pictograms (Chinese characters) to represent meaning. Chinese is also a tonal language with four pitched tones and one neutral tone. It commonly displays a basic SVO word order and, instead of conjugating the verbs to express tenses, uses aspect particles such as 了 (le) and 着(zhe) to indicate the temporal location of the sentence.

**Indonesian**  Indonesian (ind) is the national language of Indonesia (Indonesia, 2002). It is spoken by around 300 million speakers worldwide. Indonesian is developed from the literary 'Classical Malay' of the Riau-Johor sultanate (Sneddon, 2003) and has many regional variants. Indonesian is written in Latin script with a lexical similarity of over 80% to Standard Malay. Indonesian is non-tonal and has 19 consonants, 6 vowels, and 3 diphthongs. The stress is on the penultimate syllable and the word order is SVO. It has three optional noun classifiers. Indonesian has two social registers and a rich affixation system, including a variety of prefixes, suffixes, circumfixes, and reduplication. Most of the affixes in Indonesian are derivational (Pisceldo et al., 2008).

**Standard Malay**  Standard Malay (msa) is the national language of Malaysia, Brunei, and Singapore, and the language is spoken by approximately 290 million speakers worldwide. The word order of Standard Malay is SVO with four types of affixes, i.e., prefixes (awalan), suffixes (akhiran), circumfixes (apitan), and infixes (sisipan). Even though Standard Malay and Indonesian originate from the same Malay language and are mutually intelligible, they can differ in spelling and vocabulary. One example is loanwords. Due to the different colonial influences from the Dutch and British, Indonesian primarily absorbs Dutch loanwords whereas Malay absorbs English loanwords. Both languages can also differ in the meanings of the same written words, which are commonly referred to as interlingual homographs. For instance, "polisi" means "police" in Indonesian but "policy" in Standard Malay.

**Tagalog**  Tagalog (tgl) is an Austronesian language spoken in the Philippines by around 82 million native speakers. It is both agglutinative and pitch-accented, giving it rich and complex morphology (Kroeger, 1993). Tagalog's standardized form, known as *Filipino*, is the country's official national language. The difference between Filipino and Tagalog is more sociopolitical than sociolinguistic: Commonwealth Act No. 184 of 1936 created a national committee whose purpose is to "develop a national language." This resulted in the standardization of the Tagalog language into Filipino. In practice, Filipino is indistinguishable from Tagalog, albeit with the addition of letters f, j, c, x, and z, plus loanwords (Commonwealth of the Philippines, 1936).

**Vietnamese**  Vietnamese (vie), the national language of Vietnam, is spoken by around 85 million people worldwide. It is a tonal language belonging to the Austroasiatic language family and uses accents to denote six distinctive tones. The sentence structure of Vietnamese displays the SVO word order, and due to heavy influence from Chinese, it also uses a rich set of classifiers that are required in the presence of quantifiers. For instance, instead of writing "bốn gà," which literally translates into "four chickens," it should be "bốn con gà" where "con" is a classifier for non-human animate things.

**Tamil**  Tamil (tam) is a Dravidian language originating from Tamil Nadu and Sri Lanka. It is spoken by the sizeable Tamil diasporas of Singapore (2.5%

of population (Singapore, 2020)) and Malaysia (9% of population (Schiffman, 1998)), which resulted from histories of trade, migration, indentured servitude, and civil unrest. Tamil is an official language of Singapore (Singapore, 2020), and the only one originating from India. Tamil is notably diglossic, which means it has a formal literary system, lacks lexically distinctive stress, and is non-rhotic (Armstrong). Tamil uses SOV sentence structure. Tamil-English code-mixing exhibits interesting linguistic phenomena such as nonce loan, wherein many nonce borrowings from English occupy objects corresponding to Tamil verbs, and vice versa (Sankoff et al., 1990).

**Singlish**  Singlish is a widely-used conversational language in Singapore. It is an English-based creole language that arose out of prolonged language contact between speakers of many different languages in the country, including Hokkien, Malay, Teochew, Cantonese, and Tamil. Singlish is spoken by around 4 million speakers, and one unique feature of the language is its heavy use of pragmatic particles borrowed from Southern Chinese dialects. One example of this is "lah," which in the sentence, "Her dress is too short lah," emphasizes the statement.

## B   HuggingFace Inference API

We use HuggingFace's Inference API to prompt multilingual LLMs since we do not have sufficient local compute to host models with hundreds of billions of parameters such as the 176B-parameter BLOOMZ model (Muennighoff et al., 2022). The text-to-text task is treated identically as a text-generation task, and we set max_new_tokens (amount of new tokens to be generated) to 100, temperature to 0.7, and repetition_penalty to 1.2.

## C   OpenAI Inference API

We use OpenAI's official API to prompt both davinci-003 and davinci-002. Specifically, we use openai.Completion.create with a maximum generation length of 128. We use the default values for all other parameters.

## D   Flan-T5-XXL Non-English Outputs

We observe that when Flan-T5-XXL generates non-English outputs, most of them are nonsensical.

Here are some of the examples and their translations.
**Indonesian**: Ini adalah sebuah udara untuk pengobatan minyak dan di sekitar kehidupan.
*Translation: This is an air for oil treatment and around life.*
**Malay**: Artificial Intelligence adalah sebuah kantor keamanan yang digunakan untuk mengidentifikasi penduduk yang memiliki anak-anak dalam diri.
*Translation: Artificial intelligence is a security office used for identifying residents who have childen inside.*
**Tagalog**:  Weather niya ang nagsimula sa pagsasagawa ng kaniyang kargahan ng panahon.
*Translation: It was his weather that started carrying out his weather load.*
**Vietnamese**: Nhà ng tài ra mt ngi dy xut trn o trng h nhng ngi ng thng u c thit v.
*Translation: The artist has created an outstanding talent in the field of talented people.*

## E   BLOOMZ's Training Language Distribution

BLOOMZ is created by finetuning the multilingual 176B-parameter language model BLOOM (Scao et al., 2022) that is pretrained on ROOTS corpus (Laurençon et al., 2022) on a collection of prompt instructions known as xP3 (Muennighoff et al., 2022). Table 1 and Table 2 show the proportion of SEA languages investigated in our paper existing in the ROOTS and xP3 datasets respectively. Even though Indonesian and Chinese are higher in proportion than Tamil, BLOOMZ code-mix better for Tamil than the former two language with around 20% performance difference.

| Languages | Percent Distribution (%) |
|---|---|
| English | 30.04 |
| Chinese (Simplified) | 16.2 |
| Vietnamese | 2.7 |
| Indonesian | 1.2 |
| Tamil | 0.2 |

Table 1: Proportion of Languages in the ROOTS corpus (Laurençon et al., 2022).

## F   Naturalness and Fluency Issues of ChatGPT's Generation

We document a non-exhaustive list of syntactic and semantic errors as well as reasons for unnaturalness

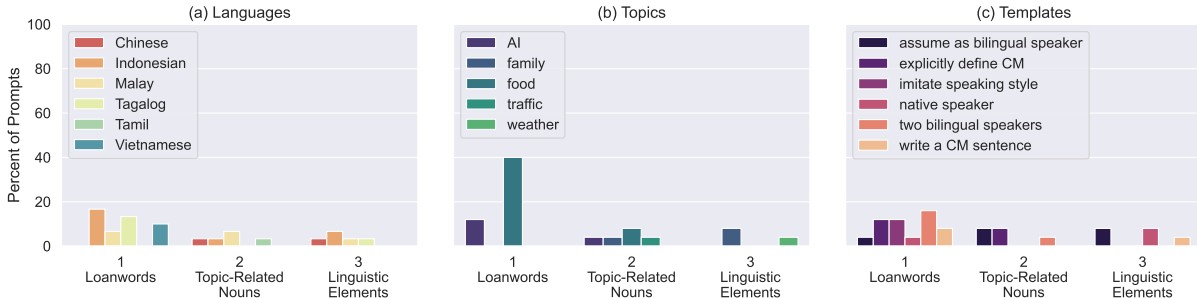

Figure 11: Analysis of davinci-002's capability of generating code-mixed data.

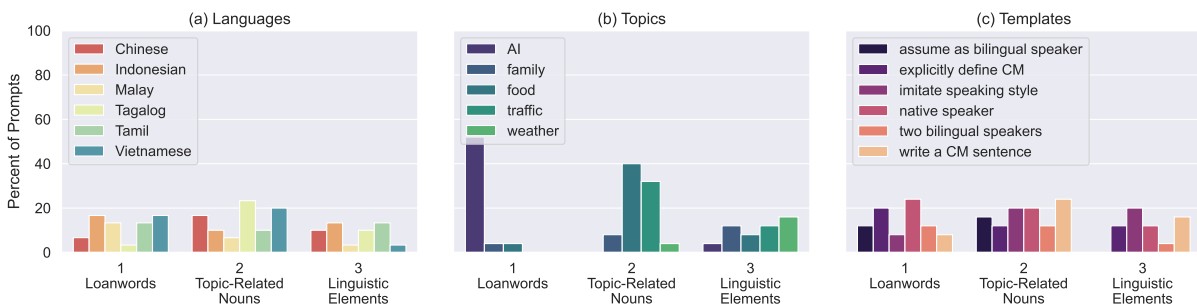

Figure 12: Analysis of davinci-003's capability of generating code-mixed data.

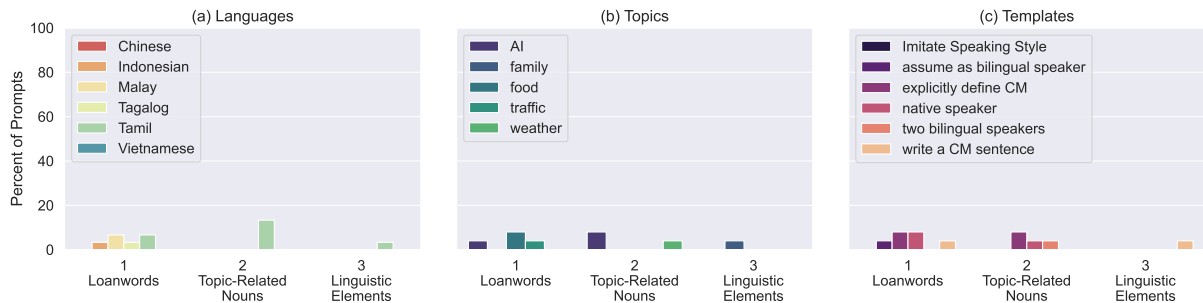

Figure 13: Analysis of BLOOMZ's capability of generating code-mixed data.

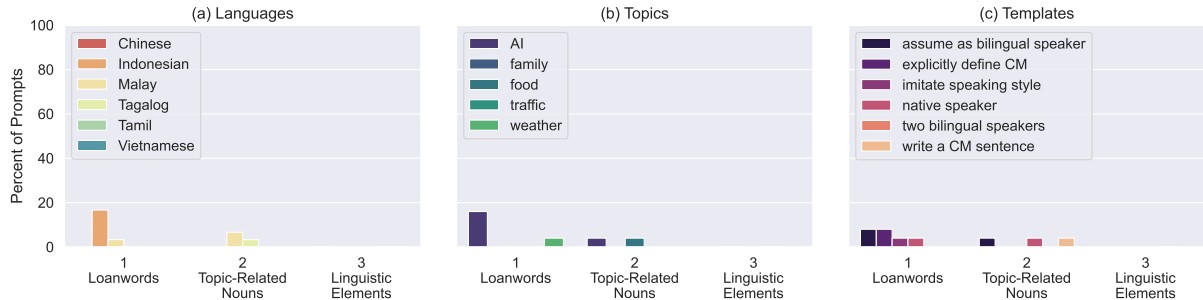

Figure 14: Analysis of Flan-T5-XXL's capability of generating code-mixed data.

| Languages | Percent Distribution (%) |
|---|---|
| English | 39.25 |
| Indonesian | 4.85 |
| Chinese (Simplified) | 4.83 |
| Vietnamese | 3.27 |
| Tamil | 0.97 |

Table 2: Proportion of Languages in the xP3 datasets (Muennighoff et al., 2022).

in ChatGPT's generation in Table 4.

# G Annotators and Inter-annotator Agreement

We have a total of 13 annotators, some of whom speak more than one SEA language. All of them are native speakers of their respective SEA languages, and most grow up in SEA. Many of our annotators are AI researchers and reside in the Global North. All the annotators are the authors of the paper.

In Table 3, we report the inter-annotator agreement scores for naturalness annotations using Fleiss' Kappa $\kappa$ (Fleiss, 1971), which measures the agreement between a fixed number of raters when assigning categorical ratings to the items. It can be applied to settings with multiple annotators and not all raters are required to annotate all items. The closer it is to 1, the higher the agreement among annotators.

According to the guideline (Landis and Koch, 1977; Altman, 1990), English-Indonesian annotations have a fair agreement, English-Chinese and Singlish have a substantial agreement, and English-Tagalog have almost perfect agreement among the annotators.

| Language | N(annotators) | $\kappa$ |
|---|---|---|
| English-Chinese | 3 | 0.6431 |
| English-Indonesian | 3 | 0.2165 |
| English-Malay | 1 | - |
| English-Tagalog | 2 | 0.8268 |
| English-Tamil | 1 | - |
| English-Vietnamese | 1 | - |
| Singlish | 3 | 0.6199 |

Table 3: Inter-annnotator agreement scores for naturalness of ChatGPT's generated code-mixed text. N(annotators) indicates the total number of annotators and $\kappa$ refers to the Fleiss' Kappa agreement score.

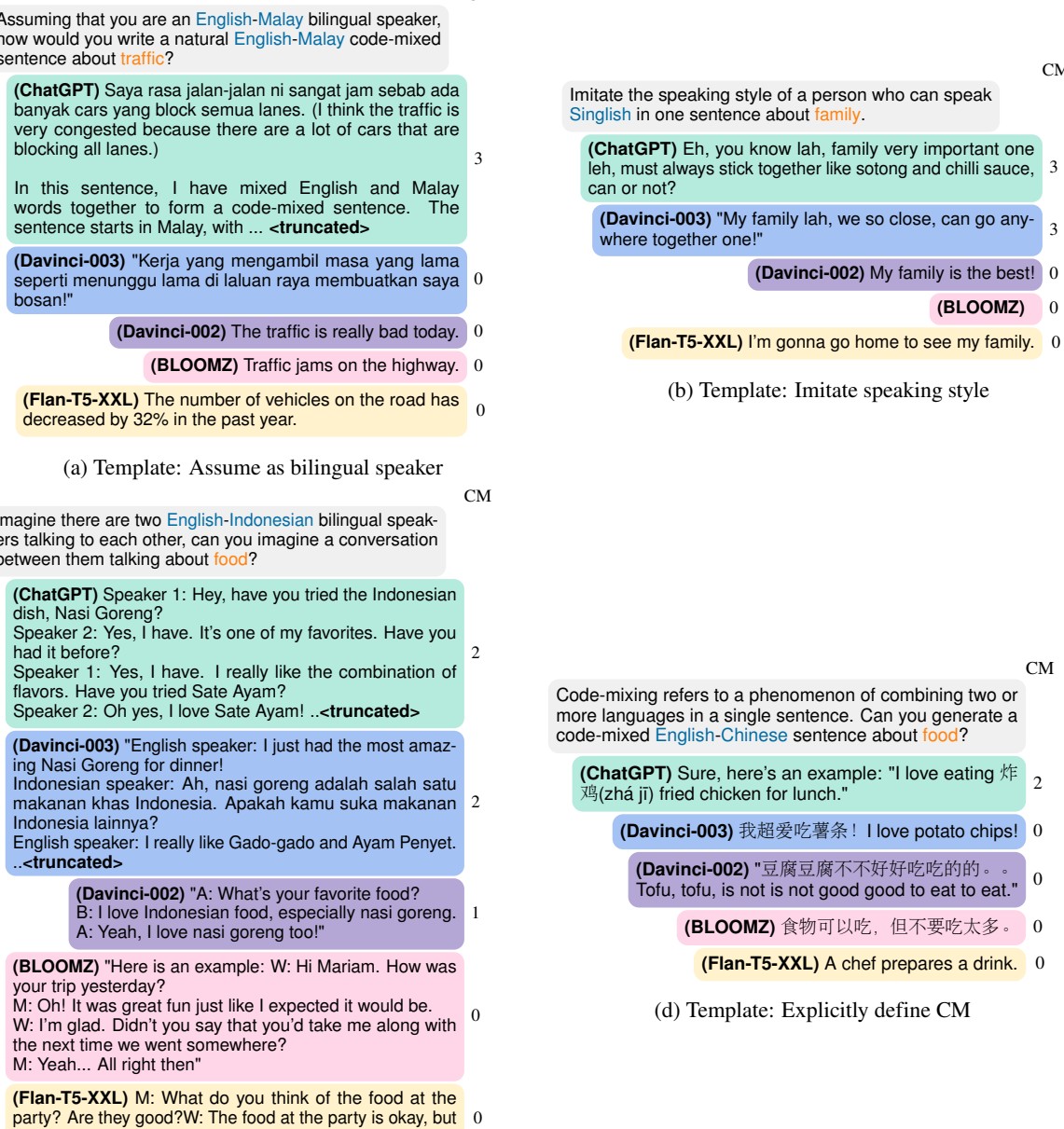

CM

Assuming that you are an English-Malay bilingual speaker, how would you write a natural English-Malay code-mixed sentence about traffic?

**(ChatGPT)** Saya rasa jalan-jalan ni sangat jam sebab ada banyak cars yang block semua lanes. (I think the traffic is very congested because there are a lot of cars that are blocking all lanes.)

In this sentence, I have mixed English and Malay words together to form a code-mixed sentence. The sentence starts in Malay, with ... **<truncated>**   3

**(Davinci-003)** "Kerja yang mengambil masa yang lama seperti menunggu lama di laluan raya membuatkan saya bosan!"   0

**(Davinci-002)** The traffic is really bad today.   0

**(BLOOMZ)** Traffic jams on the highway.   0

**(Flan-T5-XXL)** The number of vehicles on the road has decreased by 32% in the past year.   0

(a) Template: Assume as bilingual speaker

CM

Imitate the speaking style of a person who can speak Singlish in one sentence about family.

**(ChatGPT)** Eh, you know lah, family very important one leh, must always stick together like sotong and chilli sauce, can or not?   3

**(Davinci-003)** "My family lah, we so close, can go anywhere together one!"   3

**(Davinci-002)** My family is the best!   0

**(BLOOMZ)**   0

**(Flan-T5-XXL)** I'm gonna go home to see my family.   0

(b) Template: Imitate speaking style

CM

Imagine there are two English-Indonesian bilingual speakers talking to each other, can you imagine a conversation between them talking about food?

**(ChatGPT)** Speaker 1: Hey, have you tried the Indonesian dish, Nasi Goreng?
Speaker 2: Yes, I have. It's one of my favorites. Have you had it before?
Speaker 1: Yes, I have. I really like the combination of flavors. Have you tried Sate Ayam?
Speaker 2: Oh yes, I love Sate Ayam! ..**<truncated>**   2

**(Davinci-003)** "English speaker: I just had the most amazing Nasi Goreng for dinner!
Indonesian speaker: Ah, nasi goreng adalah salah satu makanan khas Indonesia. Apakah kamu suka makanan Indonesia lainnya?
English speaker: I really like Gado-gado and Ayam Penyet. ..**<truncated>**   2

**(Davinci-002)** "A: What's your favorite food?
B: I love Indonesian food, especially nasi goreng.
A: Yeah, I love nasi goreng too!"   1

**(BLOOMZ)** "Here is an example: W: Hi Mariam. How was your trip yesterday?
M: Oh! It was great fun just like I expected it would be.
W: I'm glad. Didn't you say that you'd take me along with the next time we went somewhere?
M: Yeah... All right then"   0

**(Flan-T5-XXL)** M: What do you think of the food at the party? Are they good?W: The food at the party is okay, but you don't have to eat it.   0

(c) Template: Two bilingual speakers

CM

Code-mixing refers to a phenomenon of combining two or more languages in a single sentence. Can you generate a code-mixed English-Chinese sentence about food?

**(ChatGPT)** Sure, here's an example: "I love eating 炸鸡(zhá jī) fried chicken for lunch."   2

**(Davinci-003)** 我超爱吃薯条！I love potato chips!   0

**(Davinci-002)** "豆腐豆腐不不好好吃吃的的。。 Tofu, tofu, is not is not good good to eat to eat."   0

**(BLOOMZ)** 食物可以吃，但不要吃太多。   0

**(Flan-T5-XXL)** A chef prepares a drink.   0

(d) Template: Explicitly define CM

Figure 15: All prompt templates with different languages and topic fields and responses from different LLMs containing code-mixed / non-code-mixed sentences. Note that the explanations are a part of ChatGPT's original generation. "CM" indicates the level of code-mixing (Section 2.2)

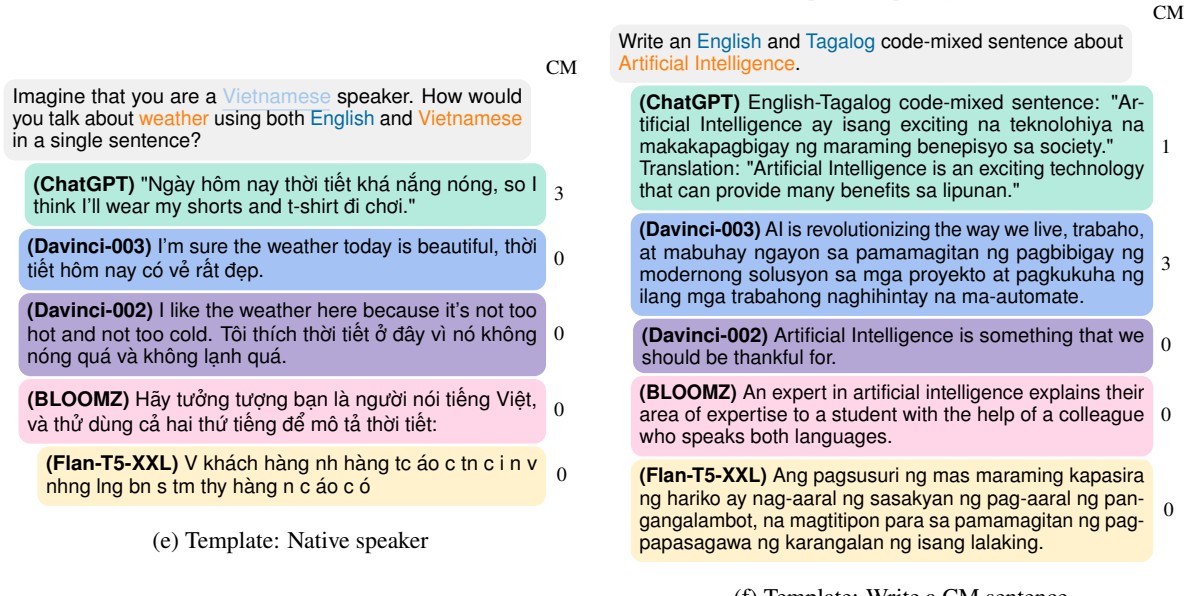

(e) Template: Native speaker

(f) Template: Write a CM sentence

Figure 15: (Continued) We also include a template where we specify the nationality of the speaker in addition to the languages and topic fields.

| Examples | Standard English Translation | Language Pairs | Issues | Explanations |
|---|---|---|---|---|
| Besok kita harus prepare payung karena cuacanya bakal cloudy dengan chance of hujan sepanjang hari | Tomorrow we need to prepare umbrella because the weather is going to be cloudy with the chance of raining all day long. | English-Indonesian | Unnatural phrasing | "chance of hujan" and "cuacanya bakal cloudy" sound unnatural. Should be "chance of raining" and "cuacanya akan menjadi cloudy" |
| Saya suka spend time bersama family saya, especially bila kita makan makanan yang sedap seperti nasi lemak or roti canai for breakfast. | I like spending time with my family, especially when we eat delicious food such as nasi lemak and roti canai. | English-Malay | Gerund; Conjunction | "suka spend time" should change to "suka spending time" as the word "suka" (like.v) should be followed with gerund. "or" should also be changed to "and". |
| I love eating 炸鸡(zhá jī) fried chicken for lunch. | I love eating fried chicken for lunch. | English-Chinese | Redundancy | "炸鸡" is the same as "fried rice". |
| So, wǒ rènwéi yī gè jiànkāng de AI xìanshí shì yī gè jùyǒu zhuǎnyè shíjì kěnéng xìng de chéngzhǎng zhìnéng de jìshù. | So, I believe a healthy AI system is a technology of growing intelligence with professional practical possibilities. | English-Chinese | Unnatural script system | The generated text should use Mandarin characters instead of Pinyin. It should be written as "So, 我认为一个健康的AI是一个具有专业实际可能性的成长智能的技术." Furthermore, the sentence does not make any sense |
| My family ay nagplano ng isang malaking family reunion sa park this coming weekend | My family is planning a big family reunion at the park this coming weekend | English-Tagalog | Possessive markers | The break from english "My family" to Tagalog "ay nagplano" is unnatural. When Tagalog is the matrix language, we use Tagalog possessive determiners, so the correct form would be "Ang family ko ay nagplano …" |
| Yesterday, tôi đã đi out với gia đình của mình để celebrate my parents' wedding anniversary. | Yesterday, I went out with my family to celebrate my parents' wedding anniversary. | English-Vietnamese | Verb phrase | Instead of "đi out", it should be either "đi" or "went out". |
| AI, you know, can do many things lah, like make our lives easier, but also can be very pai seh if we don't use it properly. | You know while AI can do many things such as making our lives easier, it can also be very embarrassing if we don't use it properly. | Singlish | Incorrect use of Singlish expression | "pai seh" is a Hokkien word that describes a person feeling shy, sorry or embarrassed. Using it to describe AI feeling embarrassed is inappropriate. |
| Eh, you know lah, family very important one leh, must always stick together like sotong and chili sauce, can or not? | Do you know that family is very important, so we must always stick together like squid and chili sauce? | Singlish | Analogy | Using "sotong and chilli sauce" (squid and chili sauce) as an analogy to familial bond is an unnatural expression. No one in Singapore uses such an expression. |
| Traffic romba kasta pattu irukku today, it's taking forever to reach my destination. | Traffic is bad today; it's taking forever to reach my destination. | English-Tamil | Adjective | "Traffic romba kasta pattu irukku today" means that the traffic is suffering, which is not the same as the traffic is congested. |
| "ஆர்டிஃபிஷியல் இணைஇயத்தள பயன்பாடுகள் வாழ்நிலையும் தகவல் அடையாளமாக எந்த மொழியிலும் பயனுள்ளதாது, Artificial Intelligence is revolutionizing the way we interact with technology." | Data identification in artificial web applications is effective in any language, Artificial Intelligence is revolutionizing the way we interact with technology. | English-Tamil | Comma splice | Both the Tamil and English independent clauses are joined by comma, which is a grammatical error of comma splice. |

Table 4: Naturalness issues with explanations for ChatGPT's code-mixed text generation.