# OpenReview forum: "Prompting Multilingual Large Language Models to Generate Code-Mixed Texts: The Case of South East Asian Languages"
_EMNLP/2023/Workshop/CALCS — EMNLP 2023 Workshop CALCS_

### Official Review · Reviewer_Zktw · 2023-10-01
**The author(s) generated a moderate amount of code-mixed data in seven languages ( Indonesian, Malay, Chinese, Tagalog, Vietnamese, Tamil, and Singlish) languages across various topics by prompting MLLMs and performed analyses and evaluations at different levels. The findings of the study are well-supported and hold significance.**

**Rating:** 4
**Confidence:** 4

**Review:**

Quality:  4

Clarity: 5

Originality:  4

Significance: 4

Pros:  The author(s) generated a moderate amount of data in seven languages ( Indonesian, Malay, Chinese, Tagalog,
Vietnamese, Tamil, and Singlish.) across various topics by prompting MLLMs and performed analyses and evaluations at different levels. The findings of the study are well-supported and hold significance.


Cons: Some of the limitations have already been mentioned in the paper.

**Candidate For Best Paper:**

Yes

**Reason For Best Paper:**

A significant contribution that provides guidelines to improve the performance of MLLM for generating code-mixed text (English-South-East Asian languages).

**Related:**

5: It is very related to the workshop.

---

### Official Review · Reviewer_4Hx5 · 2023-10-03
**Interesting data paper that investigates the quality of prompting LLMs to generate synthetic code-mixed data.**

**Rating:** 4
**Confidence:** 4

**Review:**

This paper investigates the quality of synthetic code-mixed data, generated by prompting multilingual large language models in zero-shot setup for seven languages/language-pairs: English - Indonesian/Malay/Chinese/Tagalog/Vietnamese, Tamil and Singlish. The output of the models is annotated by human annotators on different axes: degree of code-mixing (no code-mixing, loanwords, topic-related words, linguistic elements), naturalness and accurateness. The results show that publicly available instruction-tuned models (BLOOMZ, Flan-T5-XXL) are not able of producing good code-mixed data, whereas ChatGPT performs much better at the task.

## Strengths
I think the idea and data creation and evaluation is timely and interesting: there are very few manually labeled datasets for code-mixing, as for other low-resourced languages, and a lot of researchers use LLMs to construct synthetic data nowadays. It is important that researchers also evaluate the quality of these synthetically generated datasets, and list the strengths and weaknesses of the currently available models.
The paper is in general well-written and clearly structured, and the evaluation and results are presented in a clear way.
I had some problems, though, to interpret the results presented in Figure 3.
I also liked the discussion section (section 4) which elaborates on implications for LLMs and future directions to create synthetic data.\\

## Some weaknesses of the paper:
*   It would have been logical to compare the quality in a zero-shot setup with the obtained quality in a few-shot setup.
*   some of the evaluations are less interesting: e.g. mistakes made by the models when using the prompt "imagine you are a Malaysian speaker ... using both English and Chinese" leading to bad output.
*   As these synthetic data sets will be used to solve specific NLP tasks in the end, it would be nice to add evaluations for particular NLP tasks using the data sets as well.

**Candidate For Best Paper:**

No

**Reason For Best Paper:**

NA

**Related:**

5: It is very related to the workshop.